# An AR Debugging Tool for Robotics Programmers

Bryce Ikeda
University of Colorado Boulder
Colorado, United States
bryce.ikeda@colorado.edu

Daniel Szafir
University of Colorado Boulder
Colorado, United States
daniel.szafir@colorado.edu

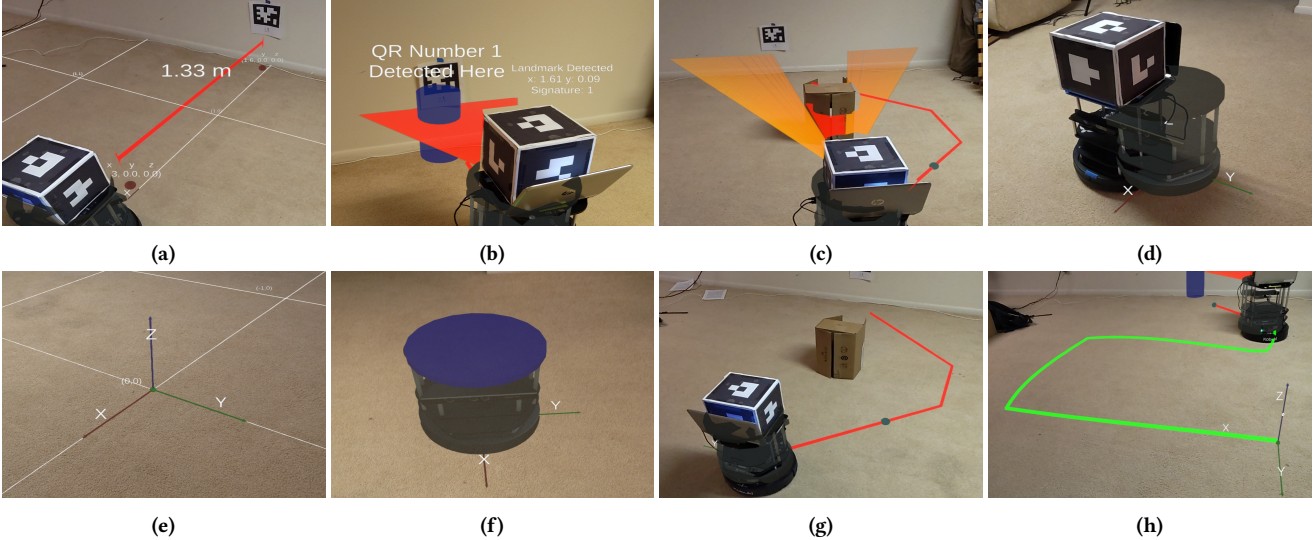

**Figure 1: In this paper we develop an AR system to support programmers debug robots. Our design can: (a) visualize and measure distance; (b) indicate detected objects and display textual information; (c) display depth sensor readings; (d) visualize the estimated position of the robot; (e) display the robot's coordinate system origin and orientation; (f) provide color indicators that can be controlled by the programmer; (g) visualize the planned path trajectory of the robot; (h) visualize the travelled path history of the robot.**

## ABSTRACT

Programming robots is a challenging task exacerbated by environmental factors, faulty hardware, and software bugs. When coding issues arise, traditional debugging techniques such as output logs or added print statements that may help in typical computer applications are not always useful for roboticists. This is because robots often have an array of sensors that output complex data types, which can be difficult to decipher as raw text data. As an alternative, we explore how an augmented reality head mounted display (ARHMD) may facilitate robotics programming by providing a medium for conveying 3D visualizations of robot data superimposed directly into the programmer's environment. Such visualizations can provide users with an intuitive way to confirm their understanding of the robot's inner state and sensor data. In this paper, we outline an augmented reality (AR) debugging tool for aiding roboticists with analyzing and fixing programs that elicit unwanted robot behavior and propose a series of planned studies to further understand the intersection of robotics programming and AR.

## CCS CONCEPTS

• **Computer systems organization** → **External interfaces for robotics**; • **Human-centered computing** → **Mixed / augmented reality**; • **User interface design**;

## KEYWORDS

Augmented Reality (AR); Mixed Reality (MR); ARHMD; interface design; robots; debugging

*VAM-HRI'21, March 8, 2021, Boulder, CO*
© 2021 Association for Computing Machinery.
ACM ISBN 978-x-xxxx-xxxx-x/YY/MM...$15.00
https://doi.org/10.1145/1122445.1122456

## 1 INTRODUCTION

Programming robots is difficult. In addition to standard challenges faced by any computer programmer, such as syntax, logic, compilation or runtime errors, roboticists must also deal with complications in added system variability due to environmental factors and irregular sensor reliability. Typical debugging techniques, such as reading raw data from print statements or log files, are both tedious and often confusing. For example, end effector transformations can

be difficult to validate via matrices, but when inspected through 3D visualizations, can be easily confirmed. To address this limitation, we are interested in leveraging the unique capabilities of AR technologies, which can project real-time data directly into the user's environment, to create an intuitive way for a user to understand the robot's operation and therefore, debug errors. AR interfaces have three key attributes that differentiate them from other systems: (1) they combine real and virtual objects in the user's current environment, (2) applications can be run interactively in real time, and (3) real and virtual objects can be aligned together and expressed as one in the scene [1].

Our work builds on research investigating the interplay between robotics and AR. For example, Collet and MacDonald examined 2D AR visualizations via their system ARDev, and found two important benefits for debugging: (1) it provides data validation preventing false conclusions, and (2) it produces immediate confirmation of hardware performance and limitations [2]. While promising, this work was limited as the visualizations were 2D and displayed on a TV screen using a top down view of the robot thus lacking stereo depth cues. This setup also requires additional perspective translation on the part of the user. In contrast, we explore the use of an ARHMD which provides all of the data visualizations within a single context in the user's real environment. We have developed our own ARHMD debugging system to both validate the findings of Collett and MacDonald and to compare the results from their 2D display with our ARHMD system to better understand the importance of *in situ* visualizations. Aspects of our AR system, such as geometric data visualizations, are also inspired by applications developed in related prior work examining AR and robotics [2–5]. We also add easily-customized visualizations for abstract data, such as textual information or color coded objects, which the programmer can take advantage of while developing their code. Going forward, we propose a small participant-observer ethnographic study of our system using two coding scenarios in which roboticists will be tasked with correcting bugs within our debugging tool. After this initial study, we plan on transitioning to a "fly on the wall" ethnographic study with a larger number of participants.

## 2  RELATED WORK

Previous studies have investigated the benefits of using AR as a tool to see the world from the robot's point of view. For example applications have been developed to help users understand the inner state of mobile robots, manipulators, and robotic swarms [2, 6, 7]. However, these systems overlay their data on 2D screens, which removes the depth cue of stereopsis and may remove the data from its true environmental context. This form of data rendering also forces the user to shift perspectives between the real-world view and the camera view. Furthermore, now that the Robot Operating System (ROS) is widely adopted for robotics applications, the software used in such previous studies is becoming obsolete. Other research has explored how AR tablets may assist K–12 students in understanding robots [3, 5, 8]. While this work helps provide insights on how to educate new roboticists, our research focuses on roboticists that are already experienced in programming ROS applications. Similar to our AR debugging tool, Muhammad et al. and Cleaver et al. have developed an AR framework for visualizing sensor data obtained

from a robot [4, 5]. Although robotic debugging is identified as a potential application for their system, their work focuses primarily on using a 2D tablet to communicate robot motion intent and improving robotic education [4, 5]. This contrasts with our system, which focuses primarily on the effects of using an ARHMD during the debugging process. ARHMDs have shown promise in many related avenues of HRI, such as providing information on robot intent, robot constraints, and sensor information for collocated users [9–12], but to our knowledge, such information has yet to be used within the context of providing assistance in robot debugging.

## 3  SYSTEM ARCHITECTURE

**ARHMD Platform:** We use the Microsoft HoloLens 2 for displaying augmented reality visualizations. The HoloLens 2 is an updated version of the HoloLens 1 with an increased field-of-view of 52 degrees and a higher display resolution of $2048 \times 1080$px. It includes multiple sensors, including an eye tracker, microphone, inertial measurement unit, depth sensor, and camera.

**Robot Platform:** Although our system is designed to be a general purpose debugging aid, for our upcoming study we will be using a TurtleBot 2. The TurtleBot is a low-cost, differential drive mobile robot commonly used as an entry machine by roboticists. The TurtleBot is controlled through a laptop running ROS, one of the leading software architectures used to program robots.

**Scene Rectification:** To provide accurately positioned visualizations, we align the coordinate frames of the TurtleBot and the HoloLens. To accomplish this, we first detect the initial position and pose of the robot by using the HoloLens camera and a fiducial marker placed on top of the TurtleBot [13]. We also use the fiducial marker to provide real-time tracking of the TurtleBot. This, in turn, provides data visualizations of the on-board depth sensor at its origin and a debugging text box on top of the robot (see Figures 1b and 1c). Once we align the origin of our space, we place a spatial anchor into our scene. Using the built-in HoloLens mapping and tracking system, the HoloLens collects data of the environment as the user moves around their workspace. Once enough data is collected, a spatial anchor can be placed as a common reference point for all visualizations. Additionally, this anchor can persist between application sessions. Therefore, the user will only need to perform a one-time environment rectification as long as the TurtleBot is placed in the same starting position between test runs.

**Data Types for Visualization:** We have developed an AR debugging tool using Unity, a 3D game engine for developing simulations and experiences, to enable data visualizations on the HoloLens. Collett and MacDonald have identified four common data types that encapsulate robot data at the interface level: scalar, vector, geometric and abstract [2]. Although they base their conclusion on data types used in Player/Stage, a different robotics platform, we determine identical information is used in ROS.

**Data Visualizations:** We represent scalar data from the depth sensor through a virtual line from the origin of the sensor along the axis of measurement for each range value (see Figure 1c). Vector data, such as the orientation of the world space, is depicted through a virtual axis game object (see Figure 1e). The red arrow points in the $x$ direction, the green arrow points in the $y$ direction and the blue arrow points in the $z$ direction. Geometric data, including

detected object locations, the TurtleBot's estimated position silhouette, the planned path trajectory and the travelled path history, can be represented through virtual objects aligned directly in the scene with their corresponding visual definitions as developed in various prior works [2–5] (see Figures 1b, 1d, 1h, and 1g). Finally, abstract data is visualized by displaying textual information in the environment and by the color change of virtual objects (see Figures 1b and 1f). To provide as little overhead as possible for the participants, scalar, vector and geometric data are displayed by default. However, abstract data can be directly altered by the text or color the participant would like to display. To send the data from ROS to our HoloLens application we use ROS#, a set of open source software libraries and tools for communicating with ROS from Unity [14].

## 4 PROPOSED EXPERIMENTAL DESIGN:

Our proposed study will be divided into two stages: (1) an informal participant-observer ethnographic pilot study, where users provide feedback on the usability of the AR debugging device and the experimental process, and (2) a larger, observer-only ethnographic study of our system where we work to answer four questions:

- In what ways can AR help with debugging, e.g., reducing actual and/or perceived debugging time, steps, effort, etc.?
- What types of software bugs is AR best suited to address?
- How might providing augmented reality data visualizations influence a programmer's debugging practice?
- While using an ARHMD, what specific aspects of augmented reality do programmers find most useful for programming a robot?

For these two stages, we are using two modified tasks proposed by Collett and MacDonald that represent standard tasks robot developers typically code: a detection task and a finder task [2]. Since these tasks require object detection and navigation, the programmer will need to take advantage of multiple forms of data types and control sequences. We will provide the user with an an autonomous robot framework that includes the code for localization, path trajectory calculations, path following control and object detection. The programmer, in turn, will only need to use our object detection output to guide their decisions for sending waypoints to our path trajectory algorithm to control the robot. The detection task is designed to be a warm-up to allow the user to become familiar with this code base. The robot will need to detect an object a few meters in front of its camera, drive up to it and stop a short distance away. Next, in the finder task, the robot should either systematically or randomly search a room until it finds a specific object. Once the robot detects the object and its position, it should drive up to it and stop a short distance away.

## 5 EXPERIMENTAL PROCEDURE

For both stages of our experimental design, we will be using the following guidelines. Participants will first be chosen from a pool of programmers with experience using ROS, but are not required to have prior experience with the HoloLens or the TurtleBot. Due to varying levels of experience and the added learning curve of using our HoloLens application, we plan for the experiment to last approximately 1–2 hours. One session will consist of five phases:

(1) introduction, (2) training, (3) task one, (4) task two, (5) conclusion. (1) Participants will be given an overview of the experiment, a consent form to sign and a pre-survey so we can determine their current experience levels with the HoloLens and programming in ROS. They will then be asked to think and process out loud while they are performing the tasks, and may be reminded if necessary [15]. This will provide us with critical insight as to the effect our AR system has on their debugging process in real time. (2) Next, the participants will be given the HoloLens and a video to watch, providing an overview of the debugging system and examples of what the two programming tasks will look like. Each participant will be required to perform multiple tasks within the virtual environment to reach a minimum experience baseline with the HoloLens. (3) Participants will begin the detection task, and will finish once the robot reaches its goal. (4) Participants will begin the finder task, and will finish once the robot reaches its goal. (5) Once participants have completed these tasks at their own pace, they will be given a post-survey based on the System Usability Scale to evaluate the usefulness of the AR debugging tool [16]. Lastly, we will conduct a short interview to gain further insight into the effects of the AR debugging tool on their coding process. A debriefing will follow.

## 6 CONCLUSION

In this paper, we describe a prototype ARHMD debugging system designed to aid roboticists in programming robots. Using the HoloLens, we provide standard visualizations for scalar, vector and geometric data, overlaying them in the user's environment in real time. In addition, we expand Collet and MacDonald's ARDev design by implementing programmer customizable visualizations for abstract data, such as text boxes and color-controlled game objects, and modernize their ideas by creating a framework for AR-supported robot debugging in ROS. We outline our proposed experimental procedure to evaluate the effectiveness of our system in assisting a participant with correctly identifying and fixing bugs in their code. We anticipate that by providing an intuitive, visual view of the inner state of the robot through *in situ* visualizations using an ARHMD, users will better understand their programs and consequently debug issues more effectively.

## ACKNOWLEDGMENTS

Thanks to Professor Thomas Howard whose template code was extended in this work.

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
