# OpenReview forum: "An AR Debugging Tool for Robotics Programmers"
_humanrobotinteraction.org/HRI/2021I/Workshop/VAM-HRI — VAM-HRI 2021 Oral_

### Official Review · AnonReviewer1 · 2021-03-04
**Review: AR Debugging Tool**

**Rating:** 8
**Confidence:** 5

**Review:**

This paper presents an ARHMD-based tool for robotics debugging, featuring various visualizations for different types of data, along with a proposed experiment for testing the tool with roboticists who may not have AR experience. The explanation of the tool and its features is thorough and provides ample detail along with clear images. The research questions address useful areas. I especially like "What software bugs is AR best suited to address?" and am curious how this will be evaluated in the study. Will you be sharing more detail about how the architecture of your system during the talk? Recommend accept - looking forward to hearing more about this fascinating research design.

---

### Decision · Program_Chairs · 2021-03-06

Accept (Oral)